# REINFORCEMENT LEARNING FOR BANDITS WITH CONTINUOUS ACTIONS AND LARGE CONTEXT SPACES

## ABSTRACT

We consider the challenging scenario of contextual bandits with continuous actions and large context spaces, e.g. images. We posit that by modifying reinforcement learning (RL) algorithms for continuous control, we can outperform hand-crafted contextual bandit algorithms for continuous actions on standard benchmark datasets, i.e. vector contexts. We demonstrate that parametric policy networks outperform recently published tree-based policies (Majzoubi et al., 2020) in both average regret and costs on held-out samples. Furthermore, we successfully demonstrate that RL algorithms can generalise contextual bandit problems with continuous actions to large context spaces and provide state-of-the-art results on image contexts. Lastly, we introduce a new contextual bandits domain with multi-dimensional continuous action space and image contexts which existing methods cannot handle.

## 1 INTRODUCTION

We consider the challenging scenario of contextual bandits with continuous actions and large "context" spaces, e.g. images. This setting arises naturally when an agent is repeatedly requested to provide a single continuous action based on observing a context only once. In this scenario, an agent acquires a context, chooses an action from a continuous action space, and receives an immediate reward based on an unknown loss function. The process is then repeated with a new context vector. The agent's goal is to learn how to act optimally. This is usually achieved via minimising regret across actions selected over a fixed number of trials.

Our work is motivated by an increasingly important application area in personalised healthcare. An agent is requested to make dosing decisions based on a patient's single 3D image scan (additional scans after treatment can potentially be damaging to the patient) (Jarrett et al., 2019). This is an unsolved domain; current methods fail to handle the large context space associated with 3D scans and the high-dimensional actions required. We consider this problem under the *contextual bandits* framework. Contextual bandits are a model for single-step decision making under uncertainty where both exploration and exploitation are required in unknown environments. They pose challenges beyond classical supervised learning, since a ground truth label is not provided with each training sample. This scenario can also be considered as one-step reinforcement learning (RL), where no transition function is available and environment interactions are considered independent and identically distributed (i.i.d).

There are well-established methods for contextual bandit problems with small, discrete action spaces, often known as *multi-armed bandits with side information*. The optimal trade-off between exploration and exploitation is well studied in this setting and formal bounds on regret have been established (Gittins et al., 1989; Auer et al., 2002; Li et al., 2010; Garivier & Moulines, 2011; Agarwal et al., 2014). However, there is relatively little research into continuous action spaces. Recent works have focused on *extreme classification* using tree-based methods to sample actions from a discretized action space with smoothing (Krishnamurthy et al., 2020; Majzoubi et al., 2020). However, developments in RL for continuous control beg the question: *Can we use one-step policy gradients to solve contextual bandits with continuous actions?*

Real world contextual bandit problems, such as those in healthcare, require solution methods to generalise to large context spaces, i.e. directly from images. We posit that existing tree-based methods are not well suited to this task in their current form (Krishnamurthy et al., 2020; Majzoubi et al., 2020; Bejjani & Courtot, 2022). However, recent breakthroughs in deep RL (Lillicrap et al., 2016; Arulkumaran et al., 2017; Schrittwieser et al., 2020; Hafner et al., 2021) have successfully demonstrated continuous control abilities directly from pixels. In this regard, neural networks have proven powerful and flexible function approximators, often employed as parametric policy and value networks. The adoption of nonlinear function approximators often reduces any theoretical guarantees of convergence, but works well in practice. Underpinning this recent work in deep RL for continuous control from pixels is the deterministic policy gradient that can be estimated much more efficiently than the usual stochastic policy gradient (Silver et al., 2014; Lillicrap et al., 2016).

The contributions of our work are as follows:

1. We modify an existing RL algorithm for continuous control and demonstrate state-of-the-art results on contextual bandit problems with continuous actions. We evaluate on four OpenML datasets across two settings: online average regret and offline held-out costs.

2. We propose a deep contextual bandit agent that can handle image-based context representations and continuous actions. To the best of our knowledge, we are the first to tackle the challenging setting of multi-dimensional continuous actions and high-dimensional context space. We demonstrate state-of-the-art performance on image contexts.

3. We propose a new challenging contextual bandits domain for multi-dimensional continuous actions and image contexts. We provide this challenging domain as a new testbed to the community, and present initial results with our RL agent.

Our new contextual bandit domain is based on a 2D game of Tanks: where two opposing tanks are situated at opposite sides of the game screen. The agent must learn control parameters to accurately fire trajectories at the enemy tank. There are three continuous action dimensions: agent $x$-location, turret angle and shot power. Example context images are provided in Figure 2.

## 2 RELATED WORKS

A naive approach to dealing with continuous actions is to simply discretise the action space (Slivkins et al., 2019). A major limitation of this approach is the curse of dimensionality: the number of possible actions increases exponentially with the number of action dimensions. This is exacerbated for tasks that require fine control of actions, as they require a correspondingly finer grained discretisation, leading to an explosion of the number of discrete actions. A simple fixed discretization of actions over the continuous space has been shown to be wasteful, and has led to adaptive discretization methods, e.g. the Zooming Algorithm (Kleinberg et al., 2008; 2019).

Building upon the discretisation approach, extreme classification algorithms were recently developed (Krishnamurthy et al., 2020; Majzoubi et al., 2020). These works use tree-based policies and introduce the idea of smoothing over actions in order to create a probability density function over the entire action space. The authors provide a computationally-tractable algorithm for large-scale experiments with continuous actions. However, their optimal performance guarantees scale inversely with the bandwidth parameter (the uniform smooth region around the discretised action). The authors of these works provide no theoretical or empirical analysis investigating the effects of large context sizes. For these reasons, we propose an RL agent based on one-step policy gradients (Sutton et al., 1999) that scales well with context size to handle continuous actions and large context spaces.

Prior works have framed personalised healthcare problems as contextual bandits problems (Kallus & Zhou, 2018; Rindtorff et al., 2019). A one-step actor-critic method was proposed for binary actions relating to just-in-time adaptive interventions (Lei et al., 2017). However, until now, these methods have been restricted to discrete actions and small context vectors.

In contrast, the notion of deep contextual bandits has previously been introduced in (Zhu & Rigotti, 2021) to handle large context spaces. The authors propose a novel sample average uncertainty method, however, it is only suitable for discrete action spaces. To the best of our knowledge, no prior works have focused on the challenging intersection of continuous actions and large context spaces, and it is an under-explored research area of particular interest in healthcare.

## 3 PRELIMINARIES

**Setting and key definitions**  We consider the i.i.d contextual bandit setting with continuous actions. In this setting, an agent acquires a context vector (or image) $x_t$ from the context space $\mathcal{X}$ by observing the environment $E$ at timestep $t$. The agent chooses an action $a_t$ from a continuous action space $\mathcal{A} = [0, 1]^N$, and receives an immediate cost based on an unknown loss function $l_t : \mathcal{X} \times \mathcal{A} \to [0, 1]$. The process is then repeated with a new context at time $t + 1$. Unlike the standard RL setting, there is no transition function in the bandit setting. Existing contextual bandit literature for continuous actions make the restriction $N = 1$, i.e. a one-dimensional action is used. This constraint can be relaxed for our proposed RL agent, but for notational simplicity, we will restrict ourselves to this case for now.

In general, an agent's behaviour is defined by a policy. We define a stochastic policy $\pi_\theta$ that maps from the context space $\mathcal{X} \to \mathcal{P}(\mathcal{A})$, where $\mathcal{P}(\mathcal{A})$ is the set of probability measures on $\mathcal{A}$, $\theta \in \mathbb{R}^n$ is a vector of $n$ parameters, and $\pi_\theta(a_t \mid x_t)$ is the conditional probability density associated with the policy. We also define a deterministic policy $\mu_\theta$ that maps contexts to specific actions in $\mathcal{A}$, similarly parameterised by $\theta$.

For tree-based policy methods (Krishnamurthy et al., 2020), a *smoothing operator* is also required: $\texttt{Smooth}_h : \mathcal{A} \to \mathcal{P}(\mathcal{A})$, that maps each action $a$ to a Uniform distribution over the interval $\{a' \in \mathcal{A} : |a - a'| \le h\} = [a - h, a + h] \cap \mathcal{A}$.

Over $T$ rounds, an agent accumulates a history of interactions. At round $t$, this experience is comprised of tuples $\{(x_i, a_i, l_t(x_i, a_i))\}_{i=1}^t$.

**Regret: Online vs Offline**  One assumption in the contextual bandit setting is the availability of an oracle to provide the loss value for a given context and action pair. Traditionally, an agent's goal is to minimise the *expected cumulative regret* across a fixed number of $T$ trials, which is equivalent to minimising the online loss: $J \triangleq \sum_{i=1}^T \mathbb{E}[l_t(x_i, a_i)]$. Note that this expectation is under both the policy and a potentially stochastic loss function, i.e. not a transition function.[1] In this online setting, an agent has access to each trial only once and no prior knowledge of the environment is assumed, and no burn-in phases is allowed.

Whilst online cumulative or average regret over a fixed number of trials is popular, it is also common to allow a dedicated learning or planning phase followed by an execution phase to exploit the learning. In these cases, *simple regret* (Bubeck et al., 2011) can also be a sensible consideration. In this offline setting a dataset of experience can be collected in advance and an agent relies on multiple passes (or epochs) over the dataset of experience followed by an evaluation on a held-out portion. We know that for enough training samples offline validation will obey the Hoeffding bound, guaranteeing that with high probability the error discrepancy between the training estimate and hold-out estimate will be small (Blum et al., 1999). Therefore, for the offline setting, we report average loss on a held-out portion of the dataset.

Methods such as UCB1 are often used to minimise online cumulative regret in discrete action settings (Auer et al., 2002; Bubeck et al., 2012), or to identify the best action in a best-arm identification setting. However, we cannot maintain statistics over the infinite actions available in the continuous setting.

**Continuous Armed Bandits**  An extension to the multi-armed bandit (MAB) setting, continuous armed bandits have an infinite number of action values available in the continuous range. Tractable algorithms, with provable bounds on cumulative regret, often make use of a Lipschitz condition[2] (Agrawal, 1995; Kleinberg, 2004). However, a simple fixed discretization of arms over the continuous space has been shown to be wasteful, and adaptive discretization has become a popular approach, e.g. the Zooming Algorithm (Kleinberg et al., 2008; 2019). Adaptive discretization has the same worst-case regret as fixed discretization, but often performs better in practice. However, the regret bounds for these algorithms rely on prior knowledge of the Lipschitz constant.

---

[1] We provide an example of a stochastic loss function, induced by stochastic action outcomes, in our new Tanks domain in Section 5.2.

[2] A function $f : X \to \mathbb{R}$ which satisfies $|f(x) - f(y)| \le L.|x - y|$ for any two arms $x, y \in X$, is called Lipschitz-continuous on X, with Lipschitz constant L.

A recent state-of-the-art method proposed smoothing over a discretised action space in order to create a density over the entire continuous action space (Krishnamurthy et al., 2020; Majzoubi et al., 2020). Continuous Actions + Trees + Smoothing (CATS) is one such algorithm that uses a pre-specified bandwidth and number of discrete action bins to perform the task of *extreme classification* for the continuous action setting (Majzoubi et al., 2020). An open-source re-implementation of the CATS algorithm is CATX (written in JAX) (Bejjani & Courtot, 2022). CATX additionally shares parameters within tree-layers using multi-class classification networks instead of binary trees. We compare our RL agent to both the CATS and CATX algorithms in Section 5.3, and describe each in more detail next.

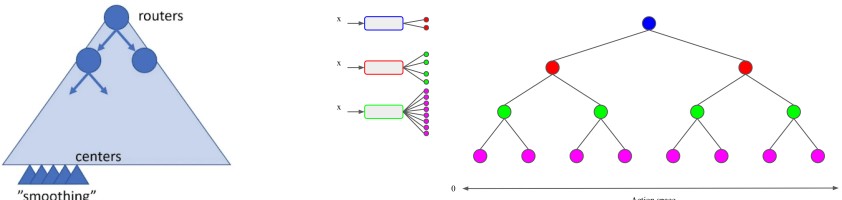

Figure 1: (Left:) CATS binary tree policy where each internal node routes a context vector left or right until a leaf node that represents an action center. Smoothing is then applied to create a probability distribution over $\mathcal{A}$, and an action is sampled. Image taken from (Vrousgou, 2021). (Right): CATX tree policy: re-implementation with multi-class classification networks at each tree-layer. Action discretization of $8$ (pink leaves), tree depth of three: blue, red and green classification networks, for input context $x$. Image taken from (Bejjani & Courtot, 2022).

**Continuous Action + Trees + Smoothing (CATS)**    The CATS algorithm (Majzoubi et al., 2020) was recently introduced as state-of-the-art for efficient contextual bandits for continuous action spaces. It is based on the idea of extreme classification using a decision tree structure where internal nodes route a context vector to a probability distribution over continuous actions. That is, the algorithm performs successive classifications within a binary tree structure, where the number of nodes in each subsequent layer increases exponentially. The authors define a Tree policy $\mathcal{T}$ as a complete binary tree of depth $D$ with $K = 2^D$ leaves, where each leaf $v$ has a label function $\texttt{label}(v) = \frac{0}{K}, \ldots, \frac{K-1}{K}$ from left to right respectively, and where each internal node $v$ maintains a binary classifier $f^v \in \mathcal{F}$ the class of binary classifiers from $\mathcal{X} \to \{\texttt{left}, \texttt{right}\}$.

Successive classifications route an input context to an output bin, or action "center". A CATS tree policy is depicted in Figure 1 (left). An action is then sampled $\epsilon$-greedily from a uniformly smooth region around the bin (defined by the bandwidth parameter). The key idea is to smooth the actions: each action $a$ is mapped to a distribution over actions using a smoothing operator introduced in Section 3. When the action space is the interval $[0, 1]$, $\mathcal{P}(\mathcal{A}) = [a - h, a + h] \cap [0, 1]$. The authors state that the loss function for smoothed actions is always well-behaved (Krishnamurthy et al., 2020). See (Majzoubi et al., 2020) for more details.

**CATX**    A recent re-implementation Bejjani & Courtot (2022) builds and improves upon the original CATS algorithm. It has been written in JAX and released open-source. The repository authors make two key contributions: $i$) they integrate nonlinear function approximators into the tree structure. Instead of binary classifiers at each internal node they implement multi-class classification networks at each tree layer; $ii$) each layer is represented by a network with output shape equivalent to the nodes in that tree-layer, meaning that each layer *shares parameters*. This is intended to stabilise training. Figure 1 (right) depicts a CATX tree policy of depth three comprising of three classification networks (blue, red and green), providing action discretisation of 8 (bins). An action is similarly sampled $\epsilon$-greedily from a smoothed $\mathcal{P}(\mathcal{A})$, i.e. a smoothed region around the leaf node.

**Deterministic Policy Gradients (DPG)**    RL for continuous control has its origins in the Deep Q-Network (DQN) (Mnih et al., 2015) algorithm that combines advances in deep learning with reinforcement learning. This seminal work adapted the standard Q-learning algorithm in order to make effective use of neural networks as function approximators. However, while DQN tackles problems with high-dimensional observation spaces, it can only handle discrete and low-dimensional action

spaces. For learning in high-dimensional and continuous action spaces, previous work combines an off-policy actor-critic approach with a deterministic policy gradient (DPG) (Silver et al., 2014). DPG maintains an actor function $\mu(s|\theta^\mu)$, parameterised by $\theta^\mu$, which specifies the current policy by deterministically mapping states to specific actions. The critic $Q(s, a)$ is learned using the Bellman equation. However, since the action space is continuous, the critic is presumed to be differentiable with respect to the action argument. This allows for a policy gradient-based learning rule, and we can approximate $\max_a Q(s, a)$ with $Q(s, \mu(s))$.

The standard REINFORCE, or vanilla actor-critic algorithms are restricted to on-policy updates (Williams, 1992; Sutton & Barto, 2018). This means that training samples are collected according to the policy currently being optimised for. These samples are then discarded and cannot be reused. However, DPG is an off-policy actor-critic approach (Degris et al., 2012) that does not require full trajectories to learn and can reuse past interactions by storing them in an experience replay buffer. This facilitates better sample efficiency by allowing a behaviour policy to collect new training samples to be decoupled from the deterministic policy being learned, providing better exploration. This allows the behaviour policy to be stochastic for exploration purposes, but the target policy remains deterministic.

Finally, combining DPG with insights from the success of DQN, the deep deterministic policy gradients (DDPG) (Lillicrap et al., 2016) algorithm demonstrated solving continuous simulated physics tasks directly from pixels. To achieve stability, the authors borrow ideas from DQN, such as a replay buffer, soft update target networks, and added a novel noise Ornstein-Uhlenbeck process to generate temporally-correlated exploration noise.

We propose a deep contextual bandit agent for continuous actions based upon the DDPG algorithm. We describe our notation and the modifications to the standard DDPG algorithm in the next section.

## 4 METHODOLOGY

An RL problem can be specified by a Markov decision process (MDP) (Kolobov, 2012; Sutton & Barto, 2018), with state space $\mathcal{S}$ and reward function $r : \mathcal{S} \times \mathcal{A} \to \mathbb{R}$. We consider the state space equivalent to the context space $\mathcal{X}$ defined in Section 3.

In the RL setting, the value of a state can be defined as the expected total discounted future reward from that state onwards. The return is defined to be the total discounted reward from time-step $\tau$ onwards, $R_\tau^\gamma = \sum_{k=\tau}^\infty \gamma^{k-\tau} r(s_k, a_k)$, for discount factor $0 < \gamma < 1$. The agent's goal is to maximise the immediate reward plus the estimated return. Notice that $\tau$ refers to the timestep within a trajectory evolving under a policy and transition function up to some infinite horizon. Whereas, $t$ specifies the trial from $T$ total trials in a contextual bandit setting, i.e. the agent is provided with i.i.d context vectors from the environment, i.e. $x_t \sim E$. Given the absence of a transition function in the bandit setting we are not required to estimate the return $R_\tau^\gamma$, as the value of any future states collapses to the instant reward achieved at the first time point. We can therefore consider this as a one-step RL problem. Instead of maximising rewards, in the contextual bandit setting it is common to minimise the equivalent loss $l_t(x_t, a_t)$.

We can therefore define the optimal one-step action-value function based on the Bellman equation as: $Q^*(x_t, a_t) = \mathbb{E}\left[r(x_t, a_t)\right]$, and the mean-squared Bellman error (MSBE) function collapses to: $L(\phi, E) = \underset{(x_t, a_t, r_t) \sim E}{\mathbb{E}}\left[(Q_\phi(x_t, a_t) - r_t)^2\right]$, for network parameters $\phi$ and environment $E$ (Sutton & Barto, 2018). Note we can interchange the rewards $r$ with losses depending on the environment.

One can note that we no longer require an estimate of the action-value for the future states along our trajectory. A key innovation in DQN (Mnih et al., 2015) was to introduce a target Q network to decouple the Q being optimised and the Q in the temporal difference update. However, since the contextual bandit setting has no transition function, there is no need to estimate future action-values. This means that our computation requirements are reduced as we do not need to maintain inefficient target networks, or interleave learning with target-updates.

It is particularly useful to estimate the policy gradient off-policy from actions sampled using a distinct behaviour (or exploration) policy, i.e. $\beta(a_t \mid x_t) \neq \pi(a_t \mid x_t)$. We use an off-policy actor-critic algorithm (Degris et al., 2012), coupled with a deterministic policy network and deterministic policy

gradients (Silver et al., 2014). That is, we estimate the action-value function using a differentiable function approximator, and then update a deterministic parametric policy $\mu_\theta$ in the direction of the approximate action-value gradient.

The fundamental result underlying our approach is the policy gradient theorem (Sutton et al., 1999). We are specifically interested in the deterministic policy gradient (Silver et al., 2014), and we adapt it here for the contextual bandit setting:

$$
\begin{aligned}
\nabla_\theta J_\beta(\mu_\theta) &= \int_{\mathcal{X}} \rho^\beta(x) \nabla_\theta \mu_\theta(a \mid x) Q^\mu(x, a) dx, \\
&= \mathbb{E}_{x_t \sim E}[\nabla_\theta \mu_\theta(x_t) \nabla_a Q^\mu(x_t, a_t) \mid_{a_t = \mu_\theta(x_t)}],
\end{aligned}
\tag{1}
$$

where, $\rho^\beta(x)$ defines the state distribution visited under our behaviour policy $\beta$, which in the bandit setting is equivalent to sampling context from our environment $E$.

There is a crucial difference between the stochastic policy gradient and the deterministic policy gradient: In the stochastic case, the policy gradient integrates over both state and action spaces, whereas in the deterministic case, it only integrates over the state space. This has additional benefits for continuous or high-dimensional action spaces.

For exploration, we take inspiration from the literature (Lillicrap et al., 2016). We construct a behaviour policy $\beta$ by adding noise sampled from a noise process $\mathcal{N}$ to our deterministic actor policy:

$$
\beta = \mu(x_t \mid \theta_t^\mu) + \mathcal{N},
\tag{2}
$$

where $\mathcal{N}$ is chosen simply as a one-step Ornstein-Uhlenbeck process to add exploration.

## 5 EXPERIMENTS

### 5.1 STAGE 1: CONTEXTUAL BANDIT WITH CONTINUOUS ACTIONS FROM VECTOR CONTEXTS

Following the experimental protocol provided in (Majzoubi et al., 2020) and (Bietti et al., 2021), we first evaluate the performance of our RL agent on four benchmark OpenML datasets (Vanschoren et al., 2013) each with a single continuous action dimension and vector context space:

1. Wisconsin dataset: 32 dim state vector, 1 action dim, 194 samples
2. CPU_act dataset: 21 dim state vector, 1 action dim, 8192 samples
3. Zurich Delays $5\%$ dataset: 17 dim state vector, 1 action dim, 26,670 samples
4. Black friday dataset : 9 dim state vector, 1 action dim, 166,821 samples

We demonstrate the performance of our RL agent compared to state-of-the-art tree-based policy method CATS (Majzoubi et al., 2020), and the shared parameter JAX re-implementation CATX (Bejjani & Courtot, 2022) as described in Section 3.

We used an implementation of CATS released with publication (Majzoubi et al., 2020). It is developed in the open-source Vowpal Wabbit framework (Langford et al., 2022). The code performs a parameter sweep over discretization and smoothing parameters: $\mathcal{J} = \{ (h, K) : h \in \{2^{-13}, \ldots 2^{-1}\}, K \in \{2^2, \ldots 2^{13}\}\}$, and we report the best results. The hyperparamer settings for all experiments can be found in Appendix A.1.1. Similarly, we used the open-source JAX implementation of CATX code (Bejjani & Courtot, 2022), and also swept for optimal hyperparameters per dataset.

Following the literature, we also compare to two baseline approaches: Firstly, $dLinear$, a discretized $\epsilon-$greedy algorithm which by default uses a doubly robust approach for policy evaluation and optimisation (Si et al., 2020). Secondly, $dTree$, is a discretized tree-based algorithm which is equivalent to CATS without smoothing, i.e. zero bandwidth. For all experiments we used $\epsilon = 0.05$.

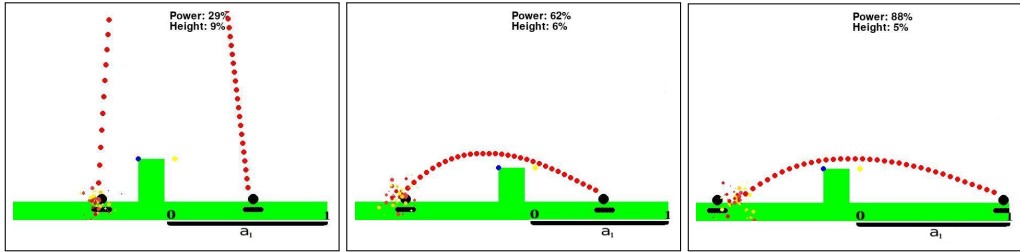

Figure 2: Three example images from our new Tanks Bandit domain. Demonstrated here are three successful actions. Tanks in black (enemy is left, agent is right of barrier), trajectory of action taken shown in red. 3-dimensional continuous actions are interpreted as $a_1$ : $x$-location, $a_2$ : shot power, $a_3$ : turret height.

## 5.2 STAGE 2: CONTEXTUAL BANDITS WITH CONTINUOUS ACTIONS FROM IMAGE CONTEXTS

We significantly increase the complexity of context provided in this section. That is, we move away from small vector context spaces, to high-dimensional contexts directly from pixels. This demonstrates the agent's ability to adequately represent the context information provided for continuous control. We experiment with two image domains: 1) a single-action regression task based on the widely used MNIST dataset (LeCun et al., 1998); and 2) we introduce a novel image benchmark domain for multi-dimensional actions based on a 2D game of Tanks. See Figure 2 for sample game images. We provide an OpenAI Gym (Brockman et al., 2016) interface to existing source game (Flair, 2022).

MNIST contains 60K training and 10K held-out samples, where each is a $28 \times 28$ image. We define a continuous loss function $l(x_t, a_t) = |a_t - y_t|$, where $y_t$ is the label provided as supervision (i.e. the digit value).

The Tanks domain is a new benchmark that we will release with the paper. It has two challenging properties over existing contextual bandit benchmark datasets: 1. large image context space (upto $800 \times 600$ pixels); and 2. multi-dimensional continuous action space, $\mathcal{A} = [0,1]^N$, where $N = 3$. The context $x_t$ can be provided as either a vector describing the locations of key objects in the scene, or as an image.

Context images contain the ground, a random barrier, and an enemy tank (randomly positioned to the left of the barrier). A single continuous action vector is available to the agent: $a_1$ : the $x$-location of its own tank (restricted to right of the barrier); $a_2$ : the power of its shot; and $a_3$ : the height of its turret.[3] Resulting trajectories can be see in Figure 2, (note these images contain additional information than the context images).

We have developed two loss functions. The first is a smooth loss across the enemy's possible locations, defined as:

$$l_t(x, a) = \begin{cases} |x_{\texttt{intersect}} - x_{\texttt{enemy}}|/x_{\texttt{barrier}} & \text{if } (0 - \delta) \le x_{\texttt{intersect}} \le (x_{\texttt{barrier}}) \\ 1 & \text{otherwise,} \end{cases} \quad (3)$$

where $x_{\texttt{intersect}}, x_{\texttt{enemy}}$ and $x_{\texttt{barrier}}$ are the $x$-location where the fired trajectory intersects with the ground plane; the $x$-location of the enemy tank; and the $x$-location of the random barrier respectively. $\delta$ is set to 50 pixels by default.

The second is a sparse loss, defined only at the enemy's location:

$$l_t(x, a) = \begin{cases} 0 & \text{if } |x_{\texttt{intersect}} - x_{\texttt{enemy}}| \le 30 \text{ pixels} \\ 1 & \text{otherwise,} \end{cases} \quad (4)$$

Each agent was trained for 50 epochs and 50 batches per epoch (further details in Appendix B).

---

[3]In Figure 2 the Power action is scaled $[0, 1] \to [0, 100]$ and turret height $[0, 1] \to [0, 9]$ for visual purposes.

## 5.3 RESULTS

To evaluate our agents we provide metrics based upon the two settings introduced in Section 3: online regret minimisation, and offline held-out costs. Online regret is popular in the Bandits literature, where the agent has access to each training sample only once. Offline costs is popular in the RL literature, and facilitates multiple passes over the dataset of experience. We simulate this setting by passing over the training dataset for multiple epochs, and report the average cost on a held-out validation set ($10\%$ of the dataset size unless otherwise stated).

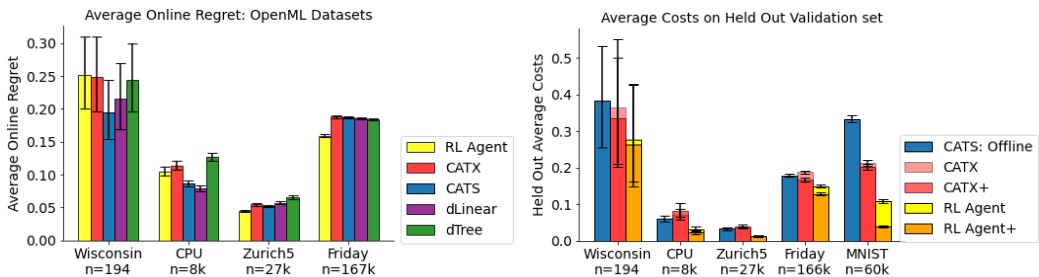

Figure 3: (Left:) Average regret in the online setting on four benchmark OpenML datasets with single continuous action. Dataset size and number of fixed trials $T$ is provided along the $x$-axis. (Right:) Average cost on a held-out validation set ($10\%$ unless stated). All confidence intervals are calculated with a single run using the Clopper-Pearson interval with $95\%$ confidence level (as is standard in the literature). The confidence intervals are noticeably larger for the small dataset sizes: Wisconsin and CPU datasets.

**Online Regret**  We report online regret on four benchmark OpenML datasets with vector contexts in Figure 3 (left) (where lower is better). We provide the dataset size, and therefore the number of fixed trials $T$, reported along the $x$-axis (increasing from left to right). The results shows a clear trend: as the number of online trials increases, i.e. for larger datasets, our RL agent begins to noticeably outperform the tree-based methods in average online regret. For example, Wisconsin is a very small dataset, and provides just 194 online trials to the agent. In this case, the tree-based method CATS achieves the lowest online regret. However, as the number of trials increases, for example in the Zurich or Friday datasets, our RL agent achieves substantially lower regret. The RL agent's policy and value networks require a certain number of trials and policy gradient updates to converge, after initialising with random weights. This experiment demonstrates that with increasing number of trials, our RL agent provides lower average regret.

**Offline Costs**  We report average offline costs on the four OpenML datasets in Figure 3 (right) (lower is better). We hold out $10\%$ of the available dataset and report average costs on the held-out samples. In Figure 3 (right) we shaded the additional improvement in performance by re-using the training data and iterating between epoch 0 and epoch 10 (30 for Wisconsin dataset), and depicted the methods using "+" in the legend. The results demonstrate a clear trend, that on held-out samples our RL agent outperforms the tree-based methods on all datasets. The shared-parameter implementation CATX also outperforms CATS. This experiment demonstrates that on held-out samples, our RL agent better generalises to new context vectors and outperforms current methods.

**Image Context**  Held-out costs on the MNIST domain are reported in Figure 3 (right) (where lower is better). We can clearly see our RL agent achieves the lowest average cost, and continues to improve with more epochs of training. We speculate that our actor-critic method is able to learn the required representations of the high-dimensional contexts to provide accurate continuous actions without discretisation. Further, the shared-parameter CATX implementation outperforms the original CATS that is clearly not able to adequately route image contexts to continuous actions.

We provide the initial results applying our RL agent to the Tanks domain with multi-dimensional continuous actions in Figure 4 (lower is better). We present online regret (left) and held-out costs (right) for four differnt agents: 1) from small vector contexts (blue); 2) from image contexts with

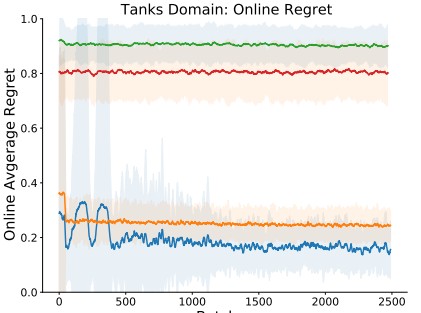 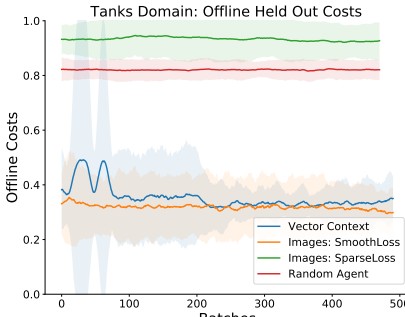

Figure 4: (Left:) Average regret in the online setting for Tanks domain. (Right:) Average cost on a held-out validation set. We compare using low-dim vector contex, image contexts with smooth and sparse loss functions, and a random agent. Confidence intervals are generated by multiple (5) training runs and each agent was trained for 50 epochs, with 50 batches per epoch.

smooth loss function in Equation 3 (orange); 3) from image contexts with challenging sparse loss function in Equation 4; 4) random agent. All images were rescaled to $(32 \times 32)$ greyscale images.

## 6 DISCUSSION

We have demonstrated that one-step RL agents, based on deterministic policy gradients, can outperform existing methods for contextual bandit problems with continuous action spaces. Our approach is effective both from vector context spaces, as is standard in the literature, and also from large context spaces where existing methods struggle.

Historically, the design of contextual bandit algorithms has been steered towards minimising cumulative (or average) regret over a fixed number of $T$ trials, with no burn-in phase. However, in real world scenarios with image contexts, we propose designing agents, or more specifically policy networks, that minimise either simple regret or held-out costs *after* an initial training period. In this way, we can exploit the potential of policy gradients to re-use previous trials off-policy. We have successfully demonstrated that this is possible from image context spaces, and it is clear this is the problem setting we must solve to address challenges in real world personalised healthcare.

Finally, existing works based on tree policies are limited to a single action dimension. This is a major limitation to their approach. In our new Tanks domain, we successfully demonstrated control of a 3-dimensional continuous action value. Our policy network based approach can be trivially extended to output more than a single continuous action value and share parameters within the network.

## 7 CONCLUSION

In this paper we have demonstrated state-of-the-art performance on contextual bandit problems with continuous actions by modifying an RL algorithm for continuous control. We outperform hand-crafted contextual bandit algorithms for continuous actions in both online regret and held-out costs. Furthermore, we have successfully demonstrated generalising contextual bandits with multi-dimensional continuous actions to large context spaces, such as images. We provided state-of-the-art performance using RL, significantly outperforming tree-policy methods on multiple benchmarks.

Finally, we have introduced a new benchmark domain for contextual bandits with multi-dimensional continuous actions and image contexts. We provide this challenging domain to the wider research community in the hope of stimulating additional research towards solving these challenging domains, especially when using sparse loss signals.

Whilst the motivation for this work comes from personalised healthcare, we leave the application of our method to medical images for future work. Similarly, for even larger context spaces such as 3D scans, we leave integrating a pre-trained image representation or a curriculum for future work.

## 7.1 ETHICS STATEMENT

There are no conflicts of interest with regards to this work. We aim to contribute to the positive body of research aimed at tackling healthcare problems. We uphold the highest research standards and report honest, trustworthy, and transparent work to the community.

Our paper introduces a new, open-source challenge domain, Tanks. Games have long provided a testbed for research into intelligent agents, from DeepBlue to AlphaGo. It is worth noting that although tanks are depicted in this game, game agents do not resemble real world tanks, and the game does not encourage violent actions. This simplified game provides our contextual bandit agent with a control problem: learn to produce a winning trajectory by adjusting parameters of the tank (the bandit actions), based on a single image of the environment (the bandit context).

This paper is under double-blind review. For this reason, we honour confidentiality and have provided no links to Github repositories or code.

## 7.2 REPRODUCIBILITY STATEMENT

The work in this paper is entirely reproducible. There are three key components to this statement: the datasets; the code; and our new Tanks environment. We address each in turn below.

We use four widely-used benchmark datasets from OpenML. They are open-source, publicly available, and version controlled. We do not manipulate the datasets, except for downloading, normalising, and randomly splitting into train and test sets. The same is true for our use of the MNIST dataset.

We will release the code required to reproduce our results in full in a linked Github repository upon publication of the paper. This will include the values of all hyperparameters and neural network architectures. We also provide many of the key hyperparameters in the supplemental material. The code to reproduce the results on benchmark algorithms, e.g. CATS, CATX, dLinear and dTree are all available at the citations provided in the main manuscript.

Upon publication we will also provide the source code for the Tanks game, along with Python code to interact with it as an OpenAI Gym environment. We will provide the code to reproduce presented results, from both vector and image context spaces.

In conclusion, significant efforts have been made to ensure the reproducibility of this work, and that the community can build upon our research.

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

# A APPENDIX

## A.1 ADDITIONAL EXPERIMENTAL DETAILS

### A.1.1 ONLINE REGRET EXPERIMENT

A parameter sweeep was performed for each dataset, for each algorithm. The results provided in Figure 3 (left) used the following parameters and hyperparameters.

**Wisconsin dataset:** id = 191
RL Agent: Actor-critic 2 hidden layers, each fully connected with dimension = 12; Relu activations; batch size = 4; memory buffer size = $100 \times$ batch size ; learning rate = (1e-3, 1e-2) (actor/critic);

CATX: Neural Net: Haiku Network with latent dimension: 10 + [2 **(depth + 1)] nodes. ; batch size = 64; Tree depth = 5; Discretisation parameter $h = 6$;

CATS: n=512; h=1;
dTree: n=512;
dLinear: n=128;

**CPU dataset:** id = 197
RL Agent: Actor-critic 2 hidden layers, each fully connected with dimension = 256; Relu activations; batch size = 4; memory buffer size = $6000 \times$ batch size ; learning rate = (3e-4,1e-3) (actor/critic);

CATX: Neural Net: Haiku Network with latent dimension: 128 + [2 **(depth + 1)]; batch size = 16; Tree depth = 4; Discretisation parameter $h = 13$;

CATS: n=8192, h=512
dTree: n=8;
dLinear: n=8;

**Zurich05 dataset:** id = 42495
RL Agent: Actor-critic 2 hidden layers, each fully connected with dimension = 128; Relu activations; batch size = 2; memory buffer size = $1000 \times$ batch size ; learning rate = (3e-4,1e-3) (actor/critic);

CATX: Neural Net: Haiku Network with latent dimension: 128 + [2 **(depth + 1)]; batch size = 64; Tree depth = 3; Discretisation parameter $h = 13$;

CATS: n=8192, h=256
dTree: n=2;
dLinear: n=4;

**Black Friday dataset:** id = 41540
RL Agent: Actor-critic 2 hidden layers, each fully connected with dimension = 128; Relu activations; batch size = 4; memory buffer size = $10000 \times$ batch size ; learning rate = (3e-4,1e-3) (actor/critic);

CATX: Neural Net: Haiku Network with latent dimension: 128 + [2 **(depth + 1)]; batch size = 64; Tree depth = 3; Discretisation parameter $h = 11$;

CATS: n==8192, h=256
dTree: n=4;
dLinear: n=4;

### A.1.2 OFFLINE COSTS EXPERIMENT

Similarly, a parameter sweeep was performed for each dataset, for each algorithm. The results provided in Figure 3 (right) used the following parameters and hyperparameters.

**Wisconsin dataset:** id = 191
RL Agent: Actor-critic 2 hidden layers, each fully connected with dimension = 12; Relu activations; batch size = 4; memory buffer size = $100 \times$ batch size ; learning rate = (1e-3, 1e-2) (actor/critic);

CATX: Neural Net: Haiku Network with latent dimension: 10 + [2 **(depth + 1)] nodes. ; batch size = 20; Tree depth = 5; Discretisation parameter $h = 6$;

CATS: n=512; h=1;
dTree: n=512;
dLinear: n=128;

**CPU dataset:** id = 197
RL Agent: Actor-critic 2 hidden layers, each fully connected with dimension = 256; Relu activations; batch size = 2; memory buffer size = $8000 \times$ batch size ; learning rate = (3e-4,1e-3) (actor/critic);

CATX: Neural Net: Haiku Network with latent dimension: 128 + [2 **(depth + 1)]; batch size = 16; Tree depth = 4; Discretisation parameter $h = 13$;

CATS: n=8192, h=512
dTree: n=8;
dLinear: n=8;

**Zurich05 dataset:** id = 42495
RL Agent: Actor-critic 2 hidden layers, each fully connected with dimension = 128; Relu activations; batch size = 32; memory buffer size = $50000 \times$ batch size ; learning rate = (3e-4,1e-3) (actor/critic);

CATX: Neural Net: Haiku Network with latent dimension: [10, 10] + [2 **(depth + 1)]; batch size = 16; Tree depth = 6; Discretisation parameter $h = 11$;

CATS: n=8192, h=256
dTree: n=2;
dLinear: n=4;

**Black Friday dataset:** id = 41540
RL Agent: Actor-critic 2 hidden layers, each fully connected with dimension = 128; Relu activations; batch size = 4; memory buffer size = $10000 \times$ batch size ; learning rate = (3e-4,1e-3) (actor/critic); schedule = lr+=0.00001 per epoch;

CATX: Neural Net: Haiku Network with latent dimension: 128 + [2 **(depth + 1)]; batch size = 64; Tree depth = 3; Discretisation parameter $h = 11$;

CATS: n==8192, h=256
dTree: n=4;
dLinear: n=4;

**MNIST dataset:**
RL Agent: Actor-critic 2 hidden layers, each fully connected with dimension = 128; Relu activations; batch size = 32; memory buffer size = $10000 \times$ batch size ; learning rate = (3e-4,1e-3) (actor/critic);

CATX: Neural Net: Haiku Network with latent dimension: 128 + [2 **(depth + 1)]; batch size = 64; Tree depth = 2; Discretisation parameter $h = 6$;

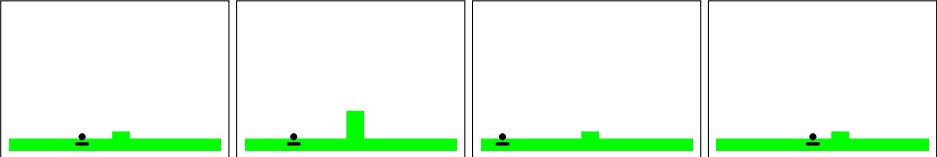

Figure 5: Three example image contexts from our new Tanks Bandit domain. Randomly generated environment, including barrier and enemy tank.

## B    TANKS DOMAIN

We provide sample context images from our new Tanks domain in Figure 5. Note, the information present in the image is the random enemy tank's location and the randomly placed barrier. The additional information in Figure 2 is for visual purposes only.

**Image RL Agent:**    Actor-critic: 1 convolutional layer: (out channels=6, kernel size=5, stride=1, padding=2), 2 fully connected layers, each fully connected with dimension = 128; Relu activations; batch size = 64; memory buffer size = 1000×batch size ; learning rate = (5e-4,1e-3) (actor/critic); schedule = lr+=0.00001 per epoch;

**Vector RL Agent:**    Actor-critic 2 hidden layers, each fully connected with dimension = 32; Relu activations; batch size = 64; memory buffer size = 1000×batch size ; learning rate = (5e-4,1e-3) (actor/critic); schedule = lr+=0.00001 per epoch;

