# OpenReview forum: "Reinforcement Learning for Bandits with Continuous Actions and Large Context Spaces"
_ICLR.cc/2023/Conference — Submitted to ICLR 2023_

### Official Review · Reviewer_rEXW · 2022-10-24

**Confidence:** 4
**Correctness:** 3
**Technical Novelty And Significance:** 1
**Empirical Novelty And Significance:** 2
**Recommendation:** 3

**Clarity, Quality, Novelty And Reproducibility:**

In formula (3), the authors have created a discontinuity in the loss function at point x=x_barrier. What purpose does it serve? Isn't a continuous function easier for learning here?

Figure 3: In the fourth paragraph of section 5.1, it seems that they have used the CATS algorithm, but in Figure 3 and the paragraph titled "Online regret" it seems that the "RL agent" is showing the result of the experiment. The right panel of Figure 3 has a different legend from the bars shown in the plot, from the legend of the right panel, and it does not include the "RL Agent." The notation needs to be more consistent.

The authors conduct an experiment on the MNIST dataset and define a continuous loss function, while the labels are discrete and it is not clear how and why they define the loss as continuous and what advantages it possesses.

The algorithm CATS [Majzoubi et al., 2020] works on continuous action spaces. So, the innovation of this work is limited to going from one to multi-dimensional action space.

Connections between the sections and subsections are not strong enough to deliver a readable story.


**Strength And Weaknesses:**

Strengths:
Extending a recent state-of-the-art method from a single dimension to a multi-dimensional continuous action space, which opens the door to new applications.
Weaknesses:
This paper introduces a new algorithm that works better than alternatives in the literature only when a large amount of data is used. This data-hungriness is a weakness, especially for a work motivated by healthcare applications. More importantly, the paper lacks any reliable analysis.


**Summary Of The Paper:**

This paper uses a deterministic policy gradient for a deep contextual bandit with additive noises for exploration, and through extensive experiments discusses that this approach is easier for continuous and multi-dimensional action spaces, compared to others.

**Summary Of The Review:**

In the end, the most important thing that the authors empirically illustrated is that for larger datasets and continuous action spaces, their method provides a smaller regret (Figure 3). It is a valuable experiment, but not a significant contribution. I will try to finalize my decision after the rebuttal period to one of the accept or reject options.

post rebuttal: the work lacks multiple things, discussed above, to become ready for publication.

---

> ### Author Response · Authors · 2022-11-10
> **Novelty is 1) extending Bandits with continuous actions to image-based contexts, *and* 2) extending to multiple action dimensions.**
>
> We thank the reviewer for their comments and remarks.
>
> Whilst it is not incorrect to state that our method outperforms the benchmarks when more data is available, that is only part of the story. We propose a novel algorithm to handle complex Bandit problems, such as those from image-based contexts and multi-dimensional continuous action spaces. In order to solve this important class of problems, more data is generally required, e.g. to train good neural network representations. In Section 5 we demonstrate state-of-the-art results on image contexts (MNIST: Figure 3 (right)), and multiple continuous action dimensions (Tanks domain: Figure 4), for  which other methods cannot handle.
>
> Furthermore, when data is scarce, we demonstrate that our RL approach outperforms existing methods on all four OpenAI benchmark datasets in the “offline” setting by reusing context-action pairs multiple times before evaluating on a held-out set. Whilst in the online setting, we outperform baselines in two of four benchmarks. We agree with the reviewer that extremely limited data is a constraint of our method, however, we argue that *previously seen data should always be available for reuse*. This means we should always be able to train for multiple epochs, improving our policy with off-policy updates. It serves no purpose to throw this data away.
>
> To clarify, in many healthcare settings data-hungriness is not prohibitive as long as we have access to either 1) a large offline dataset, or 2) a simulator to evaluate context-action pairs (as in our new Bandit Tanks domain). In either of these two cases our RL agent will have sufficient data to train neural network policy (Actor) that can generalise to new patients (contexts).
>
> Formula 3 introduces a smooth loss function with support covering the enemy’s possible locations (and loss = 1 everywhere else). The intuition is to provide zero reward (maximum loss) if the agent’s action results in the shot fired not reaching the Enemy tank’s side of the game-screen, i.e. hitting the barrier, or falling short on the agent’s side of the screen (both undesirable).
>
> Figure 3 Legend: Thank you for pointing this out. The legend on the far right of Figure 3 refers only to the three bars in Figure 3 (right) plot. As correctly pointed out, the notation is not consistent with the left plot. We have updated “DDPG” to “RL Agent” for consistency as suggested. Note that the “+” entries in the right plot correspond to only the offline setting where multiple epochs of training is performed before the held-out costs are evaluated. We have made this more clearly in the Offline Costs paragraph in Section 5.3.
>
> The MNIST continuous loss function is defined in the second paragraph in Section 5.2. We include the continuous MNIST Bandit problem as a proof-of-concept that our approach can scale to image-based contexts with a single continuous action. Figure 3 (right) demonstrates that existing baseline methods do not. However in practice, one could use different methods to solve the discrete MNIST Bandit task using discrete actions.
>
> Novelty: Whilst the CATS algorithm works on continuous action space as you stated, our novelty is twofold: 1) unlike existing works, our method extends Bandit problems with continuous actions to large image contexts **and** 2) we extend to multiple continuous action dimensions. The current state-of-the-art contextual Bandit algorithms, such as CATS, cannot handle either of these extensions as demonstrated in Section 5.2. Our RL Agent can handle both novel extensions on MNIST (Figure 3 right)), plus on our new Bandit Tanks game (Figure 4). We also demonstrate state-of-the-art performance on standard OpenML benchmark datasets in Section 5.1 (Figure 3).

---

### Official Review · Reviewer_ejwg · 2022-10-24

**Confidence:** 3
**Correctness:** 2
**Technical Novelty And Significance:** 1
**Empirical Novelty And Significance:** 2
**Recommendation:** 3

**Clarity, Quality, Novelty And Reproducibility:**

- The paper is well written except for the methodology section.
- The paper considers an important problem of contextual bandits, but the novelty of the proposed method is very limited.


**Strength And Weaknesses:**

Strengths:
- The contextual bandit problem with continuous action spaces and large contexts spaces is important and challenging.
- Related works in contextual bandits are discussed in a clear manner.
- The experimental results are provided.

Weakness:
- The description of the proposed approach is not clear and should be improved.  I am wondering whether the approach uses offline data. If so, where are offline data while related works the authors discussed are for online bandits?
- The novelty of the approach is limited: the authors adapt the DDPG algorithm with a slight modification to the contextual bandits. I am wondering why their approach with such adaptation does better than existing works which are designed for contextual bandits.
- The experimental results are not convincing. For example, the authors claim that their approach is designed for multiple-dimensional action spaces, but all experiments are for one-dimensional actions.  In addition, the number of samples of datasets is inadequate for large-scale contextual bandits.





**Summary Of The Paper:**

This paper considers a setting of contextual bandits where the space of actions is continuous and contexts are large. Different from existing approaches in contextual bandits, the paper proposes to modify the DDPG algorithm, a popular algorithm in reinforcement learning into a context bandit algorithm with continuous action spaces.  The performance of their algorithm is evaluated on benchmark OpenML datasets.  Their results show that the proposed algorithm is effective for large context spaces.





**Summary Of The Review:**

Overall, the paper can be improved by adding novelties in the approach, and experiments on large-scale datasets. I recommend rejection.

=========== after rebuttal==================
I thank the authors for their answers. However, their answers have not addressed my concerns about the novelty and reliable analyses of the proposed methodology as well as the experimental results required for high-dimensional actions. The paper raises an interesting open problem of contextual bandits and their approach is still primeval. I keep my score.

---

> ### Author Response · Authors · 2022-11-10
> **Clarifications regarding novelty**
>
> We thank the reviewer for their comments and remarks. However, we take this opportunity to clarify some of the inaccuracies in your review:
>
> In Section 3 we describe two common learning settings: i) “Online” setting where we minimise cumulative regret as is popular in Bandits literature, and ii) “Offline” setting where we report the average cost evaluated on a held-out test set as in RL literature. It would be misleading to evaluate our proposed agent on either setting in isolation. Therefore, where possible throughout the Experiments in Section 5, we provide results for **both** settings. For example in Figure 3, left refers to the online regret setting, and right relates to average offline costs. Similarly, Figure 4, left is online regret, and right is offline held-out costs. This is depicted on the y-axis of each plot, as well as in the main text.
>
> Novelty: We discuss in the Introduction and Discussion the reasons why hand-crafted Bandit algorithms for continuous actions struggle on image contexts. Recent advances in Bandit algorithms for continuous action spaces are based on extreme-classification using tree-policies. In Figure 3 (right), we demonstrate that our RL agent outperforms these tree-based methods for **image contexts**, and that only our method scales to multi-dimensional action spaces, i.e. the Tanks Bandit game (Figure 4).
>
> Please note that **not** “all experiments are for one-dimensional actions.” In Section 5.2 we introduce a new Tanks Bandit domain which uses 3-dimensional continuous action space with image contexts. The actions relate to the agent’s $x$-location, the shot power, and the turret height. We present example game scenes in Figure 2 and results in Figure 4. This is a highly novel result for Bandit problems; existing tree-based methods are incapable of handling multiple continuous actions. We demonstrate promising results with 3-dimensional continuous actions, along with challenging image-based contexts. We aim for our new Tanks domain to become a challenging new benchmark to enable future research in this area.
>
> Large-scale experiments: In addition to the contributions in the previous bullet point, Figure 3 (right) shows that our approach outperforms existing methods on four standard OpenML benchmark datasets in the offline setting where we can re-use samples. We also specifically introduce a new domain: the Tanks Bandit game, where we can evaluate as many context-action pairs as are required for the setting.

---

### Official Review · Reviewer_QaQR · 2022-10-25

**Confidence:** 3
**Correctness:** 3
**Technical Novelty And Significance:** 1
**Empirical Novelty And Significance:** 1
**Recommendation:** 3

**Clarity, Quality, Novelty And Reproducibility:**

The paper is written clearly, with good quality. While code is not available, the description of the experiments is adequate.

**Strength And Weaknesses:**

Strength:

The paper is well-organized, with each section telling exactly what it does. All the algorithms as well as the intuition behind the algorithms are explained clearly, and experimental results seem to corroborate with the claims.

Weaknesses:

1. The novelty of this paper is low. On a high-level, the paper appears to be no more than a direct application of common RL algorithms onto a special use case of RL(contextual bandits) which does not involve state change. While continuous contexts + continuous action spaces do pose their own challenges, direct application of DDPG + policy gradient based optimization is a common trick in RL.

2. While the motivation is mostly clearly stated, there seems to be a bit disconnection between the different applications/use-cases. In particular, it is unclear why MNIST is used for the problem when the graphical illustration of the problem doesn't seem to be connected with any MNIST attributes.

**Summary Of The Paper:**

This paper makes use of RL algorithms to solve the contextual bandit problems with continuous contexts/action spaces. Using DDPG and actor-critic framework in RL, the paper achieves good levels of performances as compared to other baselines.

**Summary Of The Review:**

My rating would be a 3. I'd be happy to adjust my scores if the authors address my concerns.

---

> ### Author Response · Authors · 2022-11-10
> **Novelty: We outperform state-of-the-art methods on *image contexts*, and only our method scales to multiple continuous action dimensions**
>
> We thank the reviewer for their comments and remarks.
>
> Novelty: In contrast to all existing literature, we are able to solve challenging Bandit problems with image contexts and multi-dimensional continuous actions. Whilst this may look like a “common trick”, this is highly novel work for contextual Bandit problems. This combination of challenges has not been successfully solved with any current approach.  The contextual Bandit literature often focuses on optimising online cumulative regret. We propose that this is not adequate for some of the most challenging contextual Bandit problems, e.g. directly from image contexts. Taking inspiration from RL, we propose an “offline” learning approach to contextual Bandits, i.e. we train a policy network (Actor) from interactions with a simulator. In this setting, Figure 3 (right) shows that our approach outperforms existing methods. Furthermore, in the online setting, we demonstrate improvements over existing literature in two of four OpenML baseline tasks (and competitive results for the other two).
>
> In addition to these contributions, we also demonstrate that our RL agent outperforms recently published extreme-classification methods for **image contexts**, and that only our method scales to multi-dimensional action spaces, i.e. the Tanks Bandit game. We have provided state-of-the-art results and advanced the field in an important and under-investigated domain.
>
> MNIST is a commonly used Bandits task, however, usually in the multi-armed Bandit setting with discrete action space, as in [1]. Our motivation to include a continuous MNIST Bandit task is for proof-of-concept on image-based contexts and a single continuous action. Unlike existing Bandit algorithms, our RL agent can solve Bandit problems with continuous actions **directly from image contexts**. As shown in Figure 3 (right), the recently published CATS algorithm and CATX, both fail to generalise to (28x28) image contexts. Furthermore, we introduce a new Tanks Bandit domain to evaluate our method on even larger image contexts, with **multi-dimensional continuous action space**, which existing Bandit methods are incapable of handling.
>
> Correctness: Please can you highlight which statements you are concerned with and we can respond/clarify.
>
> Reference:
> [1] G. Duran-Martin, Gerardo, A. Kara, and K. Murphy. "Efficient Online Bayesian Inference for Neural Bandits." International Conference on Artificial Intelligence and Statistics. PMLR, 2022

---

### Official Review · Reviewer_hxTb · 2022-10-27

**Confidence:** 4
**Correctness:** 1
**Technical Novelty And Significance:** 2
**Empirical Novelty And Significance:** 3
**Recommendation:** 6

**Clarity, Quality, Novelty And Reproducibility:**

Clarity - Well-written and understandable for anyone with basic bandit/RL background.

Quality - Work appears generally correct and reasonable effort was spent in comparing multiple techniques across several benchmark datasets.

Novelty - Novelty comes from applying a previously existing technique to an under-studied bandit domain. However, the insight is somewhat basic -- that bandits are a special case of RL, so we can apply RL algorithms in the bandit domain.

Reproducibility - Fully reproducible, pending open-sourcing of their code. Datasets were open source, and author even open-sourced their Open AI Gym environment "tanks".


**Strength And Weaknesses:**

Strengths:
1. Introduction of the "Tanks Bandit" OpenAI Gym interface will be a useful benchmark for the community in studying bandits with continuous output spaces.
2. Demonstrates state-of-the-art performance on several test datasets.
3. Focuses on an interesting segment of bandit problems with applications in healthcare.

Weaknesses:
1. Proposed algorithm (DDPG) is not novel. The authors merely apply it to bandits as a special case of RL. It would be interesting for the authors to explore why this connection was not made sooner. Are there other opportunities to apply well-developed RL algorithms in the bandit setting?
2. The authors compare their work to the tree-based CATS and CATX algorithms. It is not clear why Gaussian Processes are not mentioned or applied as they are also commonly used to optimize a continuous action space.
3. I have some ethical concerns about applying exploration-exploitation to determine patient dosages. It would be good to give more context on why we wouldn't always "exploit" to help the patient.
4. Unclear in figure 4 why none of the algorithms in the online setting reduce average regret over time, except perhaps transiently in the first few batches.

**Summary Of The Paper:**

In this paper, the authors study a class of contextual multi-armed bandit (MAB) problems where the input space is high-dimensional and the action space is continuous. This is a common setting in medical diagnostics where, say, a drug dosing regime must be chosen baed on medical diagnostics/imaging. The continuous action space is particularly challenging as one cannot sample the entire "arm" space. To address this challenge, they note that there has already been success in reinforcement learning (RL) for high-dimensional inputs and continuous outputs with the Deterministic Policy Gradient (DPG) algorithm. Furthermore, multi-armed bandits are just a special case of RL where we need not worry about the impact of current actions on future states. They show that a simplified version of DDPG can out-perform tree-based policies on their class of MAB problems. They also introduce a novel OpenAI Gym interface ("Tanks Bandit") as a benchmark problem.

**Summary Of The Review:**

I am marginally inclined to support this paper for publication. The material is well-presented, and focuses attention on an interesting class of bandits with high-dimensional inputs and continuous output spaces.  The paper is clear and presents a strong comparison between their DDPG-inspired technique and existing tree-based approaches. The novelty is somewhat low as they essentially re-use an existing algorithm leveraging the well-known fact that bandits are a special case of RL. However, this may inspire other applications for advanced RL techniques in the bandit domain. Also, the introduction of an OpenAI Gym interface to the tanks domain should be a valuable benchmark for this class of problems in the future.

---

> ### Author Response · Authors · 2022-11-10
> **Thank you for your thorough and positive review**
>
> We respond to each of the four points raised:
>
> 1. We postulate that no existing literature has solved the challenging domain of contextual bandits with continuous actions and image-based contexts due to the divergence of the RL and the Bandit communities. It seems each community is interested in solving a slightly different problem setting: In deep RL, it is common to have a simulator where learning large neural networks is possible and desirable, followed by a separate deployment phase where the agent is evaluated. In the Bandit community, research has mainly focused on minimising online regret bounds. Our paper bridges this gap by proposing a novel offline learning setting for Bandit problems. We provide both online and offline experiment results for completeness.
>
> 2. There is no straightforward way to apply Gaussian processes with image-based contexts. Whilst Experiments in Section 5.1 use small vector contexts, where a Gaussian process method may work well, the contribution of our paper is to develop contextual Bandit methods for *image-based contexts*. For this, GPs are ill-suited due to requiring a kernel function over high-dimensional image space. Neural networks provide us a more flexible function approximator that can generalise to new contexts, e.g. patients. Furthermore, GPs scale poorly (cubicly) with the number of datapoints.
>
> 3. We propose the “offline” Bandit setting (Introduced in Section 3) as the most appropriate for medical settings, i.e. a preliminary training phase is undertaken (exploration) to learn a policy network, followed by a deployment or “exploit” phase, as you suggest. In time, the new patient’s context can be added to the dataset, which can be used to improve the offline model at future timepoints. This offline setting is somewhat akin to planning, i.e. it doesn’t require direct interactions with the patient (or more generally, with the real world).  We do not propose taking exploratory actions on patients. However, such exploratory actions on patients would be required in the standard “online” Bandits setting to minimise cumulative regret.
>
> 4. Figure 4: The Online regret using Vector contexts (blue line), and the Image SmoothLoss (orange line) both decrease with additional batches demonstrating the models steadily improving over time. We will endeavour to display this more clearly in the figure. However, this reduction is small and it is the topic of further investigation.

---

### Decision · Program_Chairs · 2023-01-20

**Decision:**

Reject

**Justification For Why Not Higher Score:**

This paper suffers from the lack of novelty.

**Justification For Why Not Lower Score:**

N/A

**Metareview: Summary, Strengths And Weaknesses:**

This paper proposes policy gradients for solving contextual bandit problems, with a focus on continuous action and large context spaces. The proposed approach is evaluated empirically but not analyzed.

The initial reviews of the paper were 1x borderline and 3x reject, and this did not change after the discussion. The main issue is the lack of novelty, because the paper is a direct application of a popular RL technique (policy gradients) to a subclass of problems (bandits). I also skimmed through the paper and suggest the authors look at

* [Differentiable Meta-Learning of Bandit Policies](https://proceedings.neurips.cc/paper/2020/hash/171ae1bbb81475eb96287dd78565b38b-Abstract.html)

* [Meta-Learning Bandit Policies by Gradient Ascent](https://arxiv.org/abs/2006.05094)

These two papers treat policy-gradient optimization of bandit policies a bit more systematically than the reviewed paper. For instance, they also show that the optimized controllers have provably bounded regret for some conservative setting of their parameters.